# The Influence of Coumestrol on Sphingolipid Signaling Pathway and Insulin Resistance Development in Primary Rat Hepatocytes

**DOI:** 10.3390/biom11020268

**Published:** 2021-02-12

**Authors:** Hubert Zywno, Wiktor Bzdega, Adrian Kolakowski, Piotr Kurzyna, Ewa Harasim-Symbor, Klaudia Sztolsztener, Adrian Chabowski, Karolina Konstantynowicz-Nowicka

**Affiliations:** Department of Physiology, Medical University of Bialystok, 15-274 Bialystok, Poland; hubert.zywno@gmail.com (H.Z.); wbzdega@gmail.com (W.B.); adriankolakowski17@gmail.com (A.K.); pf.kurzyna@gmail.com (P.K.); eharasim@umb.edu.pl (E.H.-S.); klaudia.sztolsztener@umb.edu.pl (K.S.); adrian@umb.edu.pl (A.C.)

**Keywords:** coumestrol, sphingolipids, ceramide, insulin resistance, hepatocytes

## Abstract

Coumestrol is a phytoestrogen widely known for its anti-diabetic, anti-oxidant, and anti-inflammatory properties. Thus, it gets a lot of attention as a potential agent in the nutritional therapy of diseases such as obesity and type 2 diabetes. In our study, we evaluated whether coumestrol affects insulin resistance development via the sphingolipid signaling pathway in primary rat hepatocytes. The cells were isolated from the male Wistar rat’s liver with the use of collagenase perfusion. Next, we incubated the cells with the presence or absence of palmitic acid and/or coumestrol. Additionally, some groups were incubated with insulin. The sphingolipid concentrations were assessed by HPLC whereas the expression of all the proteins was evaluated by Western blot. Coumestrol markedly reduced the accumulation of sphingolipids, namely, ceramide and sphinganine through noticeable inhibition of the ceramide de novo synthesis pathway in insulin-resistant hepatocytes. Moreover, coumestrol augmented the expression of fatty acid transport proteins, especially FATP5 and FAT/CD36, which also were responsible for excessive sphingolipid accumulation. Furthermore, coumestrol altered the sphingolipid salvage pathway, which was observed as the excessive deposition of the sphingosine-1-phosphate and sphingosine. Our study clearly showed that coumestrol ameliorated hepatic insulin resistance in primary rat hepatocytes. Thus, we believe that our study may contribute to the discovery of novel preventive and therapeutic methods for metabolic disorders.

## 1. Introduction

The worldwide prevalence of obesity and type 2 diabetes mellitus (T2DM) have spread extensively throughout the last three decades as a consequence of a sedentary lifestyle and unhealthy diet, becoming enormous public health and social burdens [1,2]. There is no doubt that one of the most crucial factors involved in the pathogenesis of these metabolic disorders is insulin resistance. It can be defined as an impaired response to insulin in the skeletal muscles, adipose tissue, and liver. As a result, hyperinsulinemia, hyperglycemia and increased intracellular lipid concentrations in hepatocytes occur, which may finally lead to hepatic steatosis [3,4]. Elevated liver lipid concentrations are the result of (1) insulin signaling impairments, (2) intensified fatty acid (FA) transport overcoming β-oxidation capability, and (3) increased FA esterification into numerous biologically active lipid fractions (including sphingolipids), further augmenting deteriorations in hepatic insulin responsiveness. Numerous studies indicated a clear interplay between the development of insulin resistance and alterations in sphingolipid metabolism. The increased amount of saturated free fatty acids (FFA), like palmitic acid (PA), in the diet results in the promotion of the de novo synthesis of ceramide (CER), one of the most abundant and well-described sphingolipids [5,6]. Ceramide can be also generated from sphingosine through the salvage pathway or from the hydrolysis of sphingomyelin [7]. Accumulation of ceramide leads to the impairment in insulin signaling mostly by inhibiting transmission of signals through blocking Akt (protein kinase B) phosphorylation, which, in consequence, results in decreased uptake of glucose and insulin resistance development [8]. Most studies were conducted on muscle tissue, whereas there are only limited data showing the correlation between increased accumulation of various sphingolipids and the development of insulin resistance in the liver. Thus, the mechanism and possible treatment methods of these particular alterations remain partially unknown [6,9,10]. Coumestrol (COM) is a potent, naturally occurring phytoestrogen that belongs to the coumestans family and is widely found in soybeans, legumes, and brussels sprouts [11]. Like other phytoestrogens, coumestrol possesses estrogen-like activity due to its high affinity for estrogen receptors ERα and Erβ [12]. Despite the fact that coumestrol exhibits only ~2% of the binding affinity for the endogenous 17β-estradiol, it has substantially higher binding ability to the ERα and Erβ than other dietary phytoestrogens including genistein, daidzein, and mangostin [13]. Interestingly, a vast body of evidence suggests that dietary phytoestrogens, especially isoflavones, might decrease the risk of various metabolic disorders occurrence, including obesity, metabolic syndrome, and T2DM as well as possibly being effective in the adjuvant treatment [14,15,16]. Amanat et al. showed that daily supplementation with phytoestrogen from the isoflavones family, namely genistein, contributed to the amelioration of the hepatic insulin resistance among patients with non-alcoholic fatty liver disease [17]. In turn, Oza et al. revealed that treatment with another phytoestrogen that is similar in structure to the coumestrol––biochanin A––prominently reduced insulin resistance in diabetic rats [18]. None of the studies showed a possible hepatic interplay between coumestrol, sphingolipid metabolism, and insulin signaling. Thus, our study aimed to evaluate whether coumestrol via the sphingolipid metabolic pathway affects insulin signaling and, subsequently, alters hepatic insulin resistance development in the primary rat hepatocytes.

## 2. Materials and Methods

### 2.1. Animals and Study Design

This study was approved by the ethics committee on animal care at the Medical University of Bialystok. Male Wistar rats (200–250 g initial body weight, six rats in each group) were housed in approved animal-holding facilities (at 22 °C ± 2), on a reverse light–dark cycle, with unrestricted access to water and standard laboratory rat chow. Primary rat hepatocytes were isolated from rat liver by the two-step EDTA and collagenase perfusion according to the method described by Seglen [19]. Briefly, after anesthetizing the rat by intraperitoneal injection of pentobarbital in a dose of 80 mg/kg of body weight, the liver was perfused through the hepatic portal vein with Hank’s Balanced Salt Solution (HBSS, Immuniq, Zory, Poland) containing EDTA (0.1 mM). Next, the liver was treated with HBSS containing 0.05% collagenase (Sigma-Aldrich, St. Louis, MO, USA) until the disruption of intercellular junctions and destruction of the liver structure. Thereafter, the liver was removed and gently minced and obtained cells were dispersed in DMEM (Dulbecco’s Modified Eagle Medium, PAN-Biotech, Germany) containing 10% fetal bovine serum, 1% antibiotic/antimycotic, and 1% HEPES (N-2-hydroxyethylpiperazine-N-2-ethane sulfonic acid, Sigma-Aldrich, St. Louis, MO, USA). The resulting solution containing mixed cells and debris was filtered through a 100-μm filter. Subsequently, the remaining filtrate was centrifuged at 1000× *g* for 3 min at 4 °C and washed three times with DMEM. Next, the cells were separated in Percoll gradients and seeded in 12-well, collagen-coated plates. The cells were maintained in DMEM supplemented with 10% fetal bovine serum (FBS, BioWest, France) and 1% antibiotic/antimycotic (penicillin/streptomycin, Immuniq, Zory, Poland) for 24 h. The cells were cultured at 37 °C in a humidified atmosphere (5% CO_2_). After 24 h, the cells were washed twice with PBS (phosphate buffered saline, PAN-Biotech, Germany) and the morphology and viability of the attached primary hepatocytes were assessed in the Bürker chamber using Trypan blue (Sigma-Aldrich, St. Louis, MO, USA) staining during all the incubation periods. Each experiment was always carried out on 10^6^ cells per well, in which the percentage of living cells was above 85%.

### 2.2. Experimental Procedures

The treatment with palmitic acid was conducted on serum-starved cells, as previously described [20]. Before the addition of palmitate to the medium, it was conjugated with fatty acid-free bovine serum albumin (BSA, Sigma Aldrich, St. Louis, MO, USA). Briefly, palmitate stock solution was prepared by dissolving PA in a mixture of absolute ethanol and 1 M NaOH, heating to 70 °C, and conjugating with 10% BSA. Next, the stock solution was diluted in serum-free DMEM, containing 10 mM HEPES [6]. The cells were incubated in the presence or absence of palmitic acid at the concentration of 0.5 mM for 18 h. Among these two groups, half of the cells were incubated with 20 or 50 μM of coumestrol and 20 or 50 μM of COM together with 0.5 mM of PA for 18 h. The concentrations of coumestrol used in our experiment were selected based on preliminary studies, which showed that, compared to 20 μM, the 10-μM concentration did not exert statistically significant changes and 60 μM exhibited the same effects as 50 μM (see Appendix A). In another set of experiments, the hepatocytes were divided into the same experimental groups as mentioned above and were additionally chased by 100 nM insulin (NovoRapid, Novo Nordisk, ON, Canada) for the last 15 min of 18 h incubation at 37 °C. At the end of each experimental set cell morphology, viability was assessed as well (i.e., Trypan blue staining). Then, all the cells were washed three times with PBS, harvested, and homogenized in ice-cold RIPA (radioimmunoprecipitation assay buffer) lysis buffer containing protease and phosphatase inhibitors (Roche Diagnostics GmbH, Mannheim, Germany).

### 2.3. Immunoblotting Analyses

Routine Western blotting procedures were performed to detect the expression of proteins and their phosphorylated forms directly involved in the insulin signaling pathway, namely, protein kinase B [Akt, pAkt (Ser473), Cell Signaling Technology, Danvers, MA, USA] and glycogen synthase kinase 3β [GSK-3β, pGSK-3β (Ser9), Cell Signaling Technology, USA] as well as the expression of enzymes from sphingolipid signaling pathway including SPTLC1, SPTLC2 (serine palmitoyl transferase 1 and 2, Santa Cruz Biotechnology, Paso Robles, CA, USA), CerS2/4/5/6 (dihydroceramide synthase 2/4/5/6, Thermo Scientific, Rockford, IL, USA), SPHK1, SPHK2 (sphingosine kinase 1 and 2, Sigma-Aldrich, St. Louis, MO, USA), ASAH1, ASAH2 (acid ceramidase, neutral ceramidase, Santa Cruz Biotechnology, Paso Robles, CA, USA), ASAH3 (alkaline ceramidase, Sigma-Aldrich, St. Louis, MO, USA), and β-oxidation pathway enzyme β-HADH (β-hydroxyacyl-CoA dehydrogenase, Santa Cruz Biotechnology, CA, USA). Furthermore, expression of the fatty acid transport proteins FAT/CD36 (fatty acid translocase, Novus Biologicals, Cambridge), FATP2 (fatty acid transport protein 2, Santa Cruz Biotechnology, CA, USA), FATP5 (fatty acid transport protein 5, Santa Cruz Biotechnology, CA, USA) and FABPpm (fatty acid-binding protein, Abcam, UK) were measured, as it was described by Chabowski et al. [21]. Before the Western blot procedures, total protein concentration was estimated with bicinchoninic acid (BCA) assay with BSA as a standard. Next, cell lysates were reconstituted in Laemmli buffer, separated by 10% sodium dodecyl sulfate-polyacrylamide gel electrophoresis (SDS-PAGE), and transferred to nitrocellulose membranes. Then, membranes were blocked with TBS (tris-buffered saline buffer containing 5% non-fat dry milk in phosphate-buffered saline. The membranes were immunoblotted with primary antibodies of interest and incubated with appropriate secondary antibodies conjugated with horseradish peroxidase (Santa Cruz Biotechnology, CA, USA). Equal protein concentration loading was controlled by Ponceau S staining. After adding a suitable substrate for horseradish peroxidase (Bio-Rad, Hercules, CA, USA), protein bands were quantified densitometrically using a ChemiDoc visualization system EQ (Bio Rad, Warsaw, Poland). The protein expression was standardized to total protein expression, and the control group was set as 100%. The results are shown graphically as a percentage change (%) in comparison with the control group, based on six independent determinations.

### 2.4. Intracellular Sphingolipid Analyses

The content of ceramide, sphingosine (SFO), sphinganine (SFA), and sphingosine-1-phosphate (S1P) was determined as previously described in detail [22]. In brief, hepatocytes were homogenized by ultrasonication and lipids were extracted into chloroform. An aliquot of the lipid extract was transferred to a fresh tube with pre-added 40 pmol of N-palmitoyl-D-erythro-sphingosine (C17 base) (Avanti Polar Lipids) as an internal standard and then subjected to alkaline hydrolysis to deacylate ceramide to sphingosine. Free sphinganine, as well as sphingosine released from ceramide, were then converted to their o-phthalaldehyde derivatives and analyzed using an HPLC (high-performance liquid chromatography, PROSTAR, Varian Inc., Palo Alto, CA, USA) system equipped with a fluorescence detector and C18 reversed-phase column (Varian Inc. OmniSpher 5, 4.6 × 150 mm). The isocratic eluent composition of acetonitrile (Merck):water (9:1, *v*/*v*) and a flow rate of 1 mL/min was used. Before the sphingolipid analysis, the protein content was measured in all the samples with bovine serum albumin (Sigma-Aldrich, St. Louis, MO, USA) as a standard.

### 2.5. Data Analysis

The data are shown as mean ± SD based on six independent determinations (*n* = 6). The normality of the data distribution (Shapiro–Wilk test) and homogeneity of the variance (Bartlett’s test) were checked. The statistical differences of the data were performed in GraphPad Prism 5 (GraphPad Software, CA, USA) by one-way ANOVA analysis followed by an appropriate post hoc test (pairwise Student’s *t*-test). The results were considered to be statistically significant at *p* < 0.05.

## 3. Results

### 3.1. Effects of Coumestrol on Sphingolipid Concentrations in Primary Rat Hepatocytes

In primary rat hepatocytes there was a significant elevation in CER concentration among groups incubated with COM20 alone (+51.79%, *p* < 0.05, Figure 1A) and PA alone (+62.81%, *p* < 0.05, Figure 1A) compared to the control group. In contrary, the accumulation of CER was considerably decreased in hepatocytes incubated with PA + COM20 as well as PA + COM50 (−34.85%, *p* < 0.05 and −36.19%, *p* < 0.05, Figure 1A, respectively) compared with the PA group. A significant decrease in the SFO concentration was observed after incubation with PA, PA + COM20, and PA + COM50 groups (−56.22%, *p* < 0.05; −37.44%, *p* < 0.05; −54.98%, *p* < 0.05; Figure 1B, respectively) in comparison with the control group. Furthermore, the concentration of SFO notably increased after incubation with PA + COM20 compared to the PA group (+42.9%, *p* < 0.05, Figure 1B). A significant drop in SFA concentration was observed in groups treated with PA + COM20 as well as PA + COM50 (−34.57%, *p* < 0.05 and −33.56%, *p* < 0.05, Figure 1C, respectively) compared to the group incubated with PA. A significant decrease in S1P concentration was noted in either COM20 or COM50 groups incubated alone in comparison with the control group (COM20: −54.55%, *p* < 0.05; COM50: −48.18%, *p* < 0.05; PA: −64.04%, *p* < 0.05 Figure 1D) as well as in the groups incubated simultaneously with PA (PA + COM20: −33.33%, *p* < 0.05; PA + COM50: −58.38%, *p* < 0.05, Figure 1D). Additionally, the incubation with PA + COM20 resulted in substantial elevation in the S1P concentration compared to the PA group (+85.39%, *p* < 0.05, Figure 1D).

### 3.2. Effects of Coumestrol on the Expression of Insulin Signaling Pathway Proteins in Primary Rat Hepatocytes

The expression of proteins involved in the insulin signaling pathway was affected during the incubation of primary rat hepatocytes with palmitate (PA) and/or COM in the presence of insulin. We observed a significant decline in pAkt/Akt (phosphorylated protein kinase B/protein kinase B) and pGSK-3*β*/GSK-3*β* (phosphorylated glycogen synthase kinase 3*β*/glycogen synthase kinase 3*β*) expression ratios after incubation with palmitate and insulin (PA + I, 2-fold lower, *p* < 0.05, Figure 2A2; 3.6-fold lower, *p* < 0.05, Figure 2B2; respectively) compared with the C + I group. In turn, considerable elevation in expression ratio of pGSK-3*β*/GSK-3*β* was observed after incubation with PA + COM20 + I as well as PA + COM50 + I (3-fold higher, *p* < 0.05, and 2.9-fold higher, *p* < 0.05, Figure 2B2, respectively) in comparison with the PA + I group. We did not observe statistically significant results in the expression of pAkt/Akt in the groups incubated simultaneously with PA and COM. However, a trend toward increase was observed in PA + COM20 + I group (*p* = 0.09, Figure 2A2).

### 3.3. Effects of Coumestrol on the Expression of Enzymes Involved in Ceramide De Novo Synthesis Pathway in Primary Rat Hepatocytes

Incubation of primary rat hepatocytes with COM50 and PA + COM 20 as well as PA + COM50 resulted in substantial elevation of the SPTLC1 expression compared to the control group (COM50: +118.56%, *p* < 0.05; PA + COM20: +97.76%, *p* < 0.05; PA + COM50: +190.1%, *p* < 0.05, Figure 3A2). Furthermore, we noticed a significant decrease in the expression of SPTLC1 after incubation with PA + COM20 as well as PA + COM50 in comparison with the PA group (−43.7%, *p* < 0.05; −17.42%, *p* < 0.05, Figure 3A2, respectively). In turn, the expression of SPTLC2 significantly dropped in COM20 and PA + COM20 groups in comparison with the control group (−56.44%, *p* < 0.05 and −54.59%, *p* < 0.05, Figure 3B2, respectively). The expression of SPTLC2 was also decreased in the group incubated with PA + COM20 compared to the PA group (−64.54%, *p* < 0.05, Figure 3B2).

### 3.4. Effects of Coumestrol on the Expression of Enzymes That Share Ceramide De Novo Synthesis and Salvage Pathways in Primary Rat Hepatocytes

We observed notable changes in the expression of CerS6, CerS4, CerS2, and CerS5 after incubation with PA + COM20 compared to the PA group (−63.03%, *p* < 0.05 Figure 4A2; −56.74%, *p* < 0.05, Figure 4B2; −39.04%, *p* < 0.05, Figure 4C2; −20.67%, *p* < 0.05, Figure 4D2, respectively). The expression of CerS4 decreased significantly in groups incubated with COM20 and PA + COM20 in comparison with the control group (−55.1%, *p* < 0.05; −53.1%, *p* < 0.05, Figure 4B2, respectively). Furthermore, the expression of CerS5 also decreased after incubation with PA + COM50 (−17.03%, *p* < 0.05, Figure 4D2).

### 3.5. Effects of Coumestrol on the Expression of Enzymes Involved in the Sphingolipid Salvage Pathway in Primary Rat Hepatocytes

We noticed considerable increase in the expression of SPHK1 after incubation with COM50 and PA + COM20 as well as PA + COM50 groups (+74.82%, *p* < 0.05; +206.47%, *p* < 0.05; +71.48%, *p* < 0.05, Figure 5A2, respectively) compared with the control. Furthermore, the incubation with PA + COM20 resulted in increased expression of the SPHK1 and SPHK2 compared with the PA group (+75.54%, *p* < 0.05, Figure 5A2; +91.74%, *p* < 0.05, Figure 5B2; respectively). In contrary, we observed notable decline in the expression of SPHK2 after incubation with PA + COM50 compared to the control (−25.71%, *p* < 0.05, Figure 5B2). Incubation with PA + COM20 as well as PA + COM50 resulted in significantly increased ASAH1 expression in relation to the PA group (+199.35%, *p* < 0.05; +204.57, *p* < 0.05; Figure 5C2, respectively). What is more, significant elevation of ASAH3 expression was detected in PA + COM20 as well as PA + COM50 groups +105.1%, *p* < 0.05; +137.82%, *p* < 0.05; Figure 5E2, respectively) compared with the control group, whereas in primary rat hepatocytes treated with PA + COM50 we observed increase in ASAH3 expression compared with the PA group (+43.58%, *p* < 0.05, Figure 5E2).

### 3.6. Effects of Coumestrol on the Expression of Proteins Involved in Fatty Acid Uptake and β-Oxidation in Primary Rat Hepatocytes

The expression of FATP5 in the cells incubated with PA was significantly increased (+53.61%, *p* < 0.05, Figure 6A2), whereas it was decreased in groups treated with PA + COM20 (−43.95%, *p* <0.05, Figure 6A2) compared to the control. Incubation with PA + COM20 as well as PA + COM50 resulted in considerable decrease in FATP5 expression compared to the PA group (−63.5%, *p* < 0.05; −58.26%, *p* < 0.05, Figure 6A2, respectively). We did not find significant changes in the expression of FATP2 (*p* > 0.05, Figure 6B2). The expression of FABPpm markedly increased in the group incubated only with PA (+102.62%, *p* < 0.05, Figure 6C2.) as well as in groups incubated with PA + COM20 and PA + COM50 (+114.05%, *p* < 0.05; +120.19%, *p* < 0.05, Figure 6C2, respectively) compared to the control. Moreover, elevated expression of FAT/CD36 was detected in PA group (+6.05%, *p* < 0.05, Figure 6D2) and was decreased in COM20 and PA + COM20 as well as PA + COM50 groups (COM20: −41.87%, *p* < 0.05; PA + COM20: −42.15%, *p* < 0.05; PA + COM50: −55.12%, *p* < 0.05, Figure 6D2) in comparison with the control. A considerable decrease in the expression of FAT/CD36 compared to the PA group was noticed in primary rat hepatocytes treated with PA + COM20 as well as PA + COM50 (−45.45%, *p* < 0.05; −57.68%, *p* < 0.05, Figure 6D2, respectively). After treatment with coumestrol the expression of *β*-HADH was significantly decreased in COM20, PA + COM20, and PA + COM50 groups (−50.98%, *p* < 0.05; −61.58%, *p* < 0.05; −43.7%, *p* < 0.05, Figure 6E2, respectively) as well as in the group incubated only with PA (−69.27%, *p* < 0.05, Figure 6E2) compared to the control group. In turn, the expression of *β*-HADH was notably increased in PA + COM50 in comparison with the PA group (+83.23%, *p* < 0.05, Figure 6E2).

## 4. Discussion

Insulin resistance constitutes a crucial factor in the pathogenesis of various metabolic disorders such as obesity, atherosclerosis, hypertension, and type 2 diabetes mellitus. Taking into account the rapidly growing number of patients suffering from these diseases worldwide, insulin resistance is considered to be a major health threat in well-developed countries [23]. The body of evidence suggests that phytoestrogens may be efficient in the prevention and supportive treatment of diseases connected with insulin resistance. Thus, our study was focused on the assessment of phytoestrogen-coumestrol as a potential agent in the treatment of hepatic insulin resistance via its influence on sphingolipid metabolism. To show a potential influence of COM on the insulin signaling pathway, we evaluated alterations in phosphorylation of two proteins substantially involved in insulin action, namely, Akt and GSK-3β. They seem to be the most important because phosphorylation of Akt leads to phosphorylation and inactivation of GSK-3β, the enzyme that plays a pivotal role in the final step of glycogen synthesis as well as in the exacerbation of insulin resistance [24,25]. Ceramide, through stimulation of the different isoforms of the protein kinase C as well as by indirect suppression through induction of the protein phoshphatase 2, leads to the inhibition of the insulin-stimulated Akt phosphorylation [5,26]. Therefore, an excess cellular accumulation of ceramide is associated with impaired glucose uptake and, subsequently, an exacerbation of the insulin resistance. After incubation of primary rat hepatocytes with PA and COM in both concentrations (20 μM and 50 μM), we observed a substantial increase of insulin-stimulated pGSK-3β/GSK-3β expression ratio in comparison with the group incubated only with PA and insulin. These data clearly show that COM intensified the phosphorylation of GSK-3β and, subsequently, inactivation of this enzyme, which may be correlated with the beneficial anti-diabetic effect of used phytoestrogen [27]. We did not observe any significant changes in the expression ratio of pAkt/Akt in the groups treated with COM. It may be suspected that a longer incubation time with COM would be more effective in Akt phosphorylation because of the trend toward increase observed in the PA + COM20 + I group. Interestingly, Lin et al. revealed that GSK-3β activation, via its dephosphorylation, was induced by CER in the study conducted on mouse T hybridoma cell line 10I [28]. The incubation of primary rat hepatocytes with PA and COM in two different concentrations resulted in notably decreased cellular accumulation of CER and SFA. These results are in agreement with a study conducted by Vinayavekhin et al., who showed that treatment with another dietary phytoestrogen from plant *C. comosa*, namely, diarylheptanoid, decreased concentration of CER and, subsequently, reduced insulin resistance development in ovariectomized rats [29]. On the contrary, Charytoniuk et al. showed that the treatment with enterolactone, which also belongs to phytoestrogens, and PA simultaneously resulted in elevated concentration of CER in malignant hepatocytes (HepG2). However, this discrepancy might be related to the proapoptotic influence of enterolactone on the malignant HepG2 cells used in that experiment. Noteworthy, in the group incubated with COM20 alone, we reported a substantially increased concentration of CER, whereas in the study conducted by Charytoniuk et al. an increased content of SFA, not CER, after incubation with enterolactone alone was reported. This suggests that coumestrol, similarly to enterolactone, may exert various effects on the sphingolipid accumulation, depending on the presence or absence of palmitate excess and cell type [9]. As observed in our study, increased accumulation of CER may be a result of increased expression of proteins involved in FFA uptake including FABPpm, FATP, and FAT/CD36. Fatty acid transport proteins’ expression highly correlates with augmented delivery of palmitate and other FFA, which constitutes an essential substrate to the de novo CER synthesis pathway. Bonen et al. indicated that reduced FAT/CD36 expression in *CD36* knockout mice and, subsequently, diminished fatty acid transport was correlated with decreased CER accumulation and ameliorated insulin resistance occurrence in muscle cells [30]. The simultaneous incubation of primary rat hepatocytes with PA and COM in concentrations of 20 μM and 50 μM revealed notably decreased expression of both FATP5 and FAT/CD36. This might contribute to the reduced concentrations of CER observed in these groups, as a consequence of diminished delivery of the crucial substrates for the de novo synthesis of this compound. Furthermore, we noticed significantly decreased expression of the enzymes involved in the CER de novo synthesis pathway, namely, SPTLC1, SPTLC2, and CerS2/4/5/6 after treatment with combined PA and COM20. These results clearly showed that the attenuated accumulation of CER may occur through inhibition of the de novo synthesis pathway in lower COM concentration and decreased transport of PA into hepatocytes regardless of the concentration that subsequently increased insulin sensitivity. The salvage pathway, besides the de novo pathway and sphingomyelin hydrolysis, constitutes another important route of CER synthesis in the cell. Indeed, it is estimated that the sphingolipid salvage pathway may participate even in 90% of total sphingolipid biosynthesis [7]. Thus, to examine the possible involvement of this pathway, we assessed the cellular concentration of SFO and sphingosine-1-phosphate (S1P) as well as the expression of SPHK1 and SPHK2. Incubation of the cells in the presence of PA and COM20 led to notably increased expression of both sphingosine kinases (SPHK1 and SPHK2) and, as a consequence, increased accumulation of S1P in this group. Our findings are in agreement with the studies showing that lowered activity of sphingosine kinases resulted in decreased S1P concentration and impaired insulin signaling as well as the development of diabetes in [31,32]. In turn, the increased activity of these enzymes and increased S1P levels significantly ameliorated insulin sensitivity [32]. The concentration of SFO was increased in the cells incubated with PA and COM but only in a lower concentration. These changes were reflected in the increased expression of ceramidases, which transform CER into SFO. Indeed, the expression of acid ceramidase and alkaline ceramidase was significantly elevated in primary rat hepatocytes incubated with combined PA and COM simultaneously. Thus, we suspect that COM intensified S1P synthesis from SFO, a part of the salvage pathway that was intensified after treatment of primary rat hepatocytes with COM. S1P is widely known for its anti-inflammatory effect and plays a pivotal role in the function, regeneration, and survival of the cell [33]. COM may augment the synthesis of S1P in the hepatocytes, which may constitute the protective mechanism against inflammation and progression of the hepatic insulin resistance that may occur at the beginning of the steatosis development. However, some studies indicated that increased accumulation of S1P in the liver constitutes a factor of hepatic steatosis progression and development to fibrosis [34,35]. Thus, the role of S1P in the pathogenesis of hepatic insulin resistance, diabetes, and obesity in humans is still unclear and should be fulfilled by further studies. The de novo pathway and salvage pathway share the same ceramide synthases, which are crucial for generation of either dihydroceramide or ceramide. This may suggest that two, potentially very different, pathways remain linked. Thus, the cellular response on the stimulus may depend on which pathway will be promoted in the overall ceramide biosynthesis [7]. Therefore, as described above, changes in the expression of the CerS2/4/5/6 after simultaneous treatment with PA and COM may not only diminish the ceramide synthesis through de novo pathway but also by regulation of the salvage pathway. These results suggest that coumestrol led to increased synthesis of the beneficial S1P by elevating the expression of the SPHK1 and SPHK2. Moreover, it also significantly reduced the ceramide formation by decreasing the expression of CerS2/4/5/6 as these enzymes are mainly involved in the formation of ceramide from palmitate. Last but not least, Raichur et al. showed that the increased concentration of ceramides was markedly associated with impaired lipid β-oxidation and subsequently caused hepatic steatosis and insulin resistance development in CerS 2^−/+^ mice [36]. In our study, we assessed the possible influence of COM on β-oxidation by the incubation of primary rat hepatocytes with combined PA and COM50, which revealed that the expression of β-HADH was notably increased compared to the PA group, indicating that β-oxidation was enhanced. These results suggest that COM via intensified β-oxidation of FA diminishes FA availability to the de novo CER synthesis pathway. Thus, it may constitute a putative mechanism of the protection of the cell against the enhanced accumulation of CER and, in consequence, ameliorated insulin sensitivity.

## 5. Conclusions

In conclusion, our study demonstrated that the treatment of primary rat hepatocytes with coumestrol during increased availability of PA leads to the amelioration of hepatic insulin resistance. The possible mechanisms of the outcomes seem to be complex and involve alterations in the sphingolipid metabolism pathway components, including inhibiting the ceramide de novo synthesis pathway as well as augmenting the sphingolipid salvage pathway. What is more, COM significantly decreased the expression of proteins responsible for FFA uptake and enhanced the mitochondrial FA β-oxidation, which also affected the rate of sphingolipid deposition in hepatocytes. There is no doubt that the popularity of plant-based diets is increasing worldwide, being a strong alternative to Western eating patterns. Thus, we believe that our results may contribute, at least partially, to the discovery of novel preventive and therapeutic methods for metabolic disorders, especially obesity and T2DM.

## Figures and Tables

**Figure 1 biomolecules-11-00268-f001:**
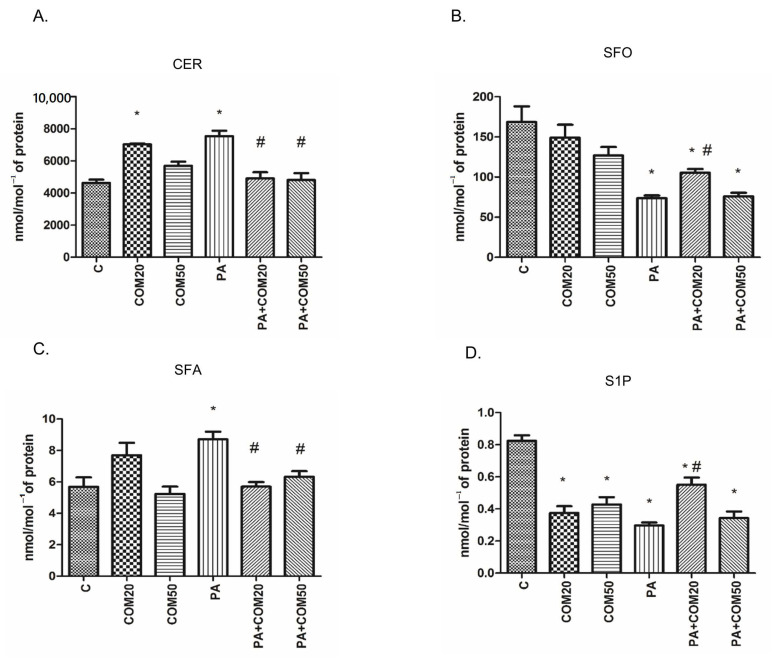
The concentrations of CER—ceramide (**A**), SFO—sphingosine (**B**), SFA—sphinganine (**C**), and S1P—sphingosine-1-phosphate (**D**) in primary rat hepatocytes. The cells were incubated with the presence or absence of PA (0.5 mM) or/and COM (20 or 50 μM), as it was described in Materials and Methods section. The concentrations of sphingolipids were measured by HPLC technique. The data are shown as the mean ± SD and are based on six independent determinations (*n* = 6); * *p* < 0.05 significant difference vs. control group; # *p* < 0.05 significant difference vs. palmitate group; C, control; COM20 or COM50, coumestrol 20 or 50 μM; PA, palmitic acid; PA + COM20 or PA + COM50, palmitic acid + coumestrol 20 or 50 μM.

**Figure 2 biomolecules-11-00268-f002:**
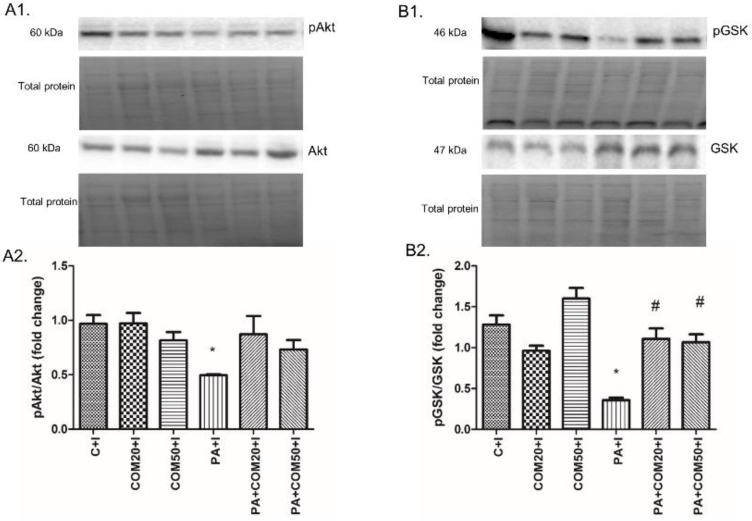
Representative Western blot membranes of evaluated proteins, i.e., pAkt/Akt—phosphorylated protein kinase B/protein kinase B (**A1**), pGSK-3β/GSK-3β—phosphorylated glycogen synthase kinase 3*β*/glycogen synthase kinase 3*β* (**B1**), and the corresponding images of total protein staining after membrane transfer used for normalization. The expression of proteins involved in insulin signaling pathway, i.e., pAkt/Akt (**A2**), pGSK-3β/GSK-3β (**B2**) in primary rat hepatocytes. The cells were incubated with presence or absence of PA (0.5 mM) or/and COM (20 or 50 μM). Additionally, every group was also incubated with insulin, as it was described in Materials and Methods section. The expression of the proteins was evaluated by the Western blot technique. The data are shown as the mean ± SD and are based on six independent determinations (*n* = 6); * *p* < 0.05 significant difference vs. control + insulin group; # *p* < 0.05 significant difference vs. palmitate + insulin group; C + I, control + insulin; COM20 + I or COM50 + I, coumestrol 20 or 50 μM + insulin; PA + I, palmitic acid + insulin; PA + COM20 + I or PA + COM50 + I, palmitic acid + coumestrol 20 or 50 μM + insulin.

**Figure 3 biomolecules-11-00268-f003:**
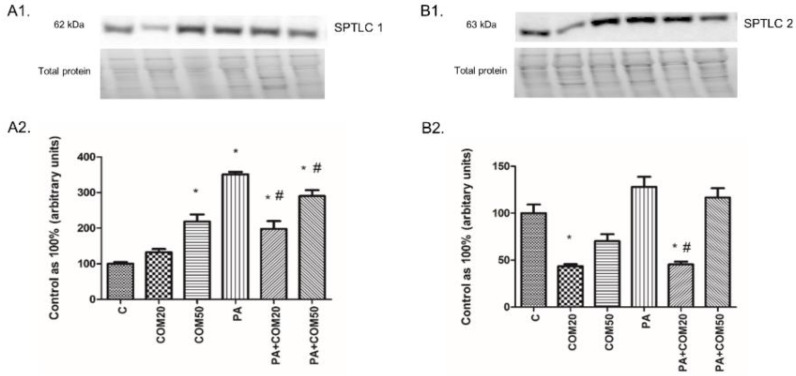
Representative Western blot membranes of evaluated proteins, i.e., SPTLC1—serine palmitoyl transferase 1 (**A1**), SPTLC2—serine palmitoyl transferase 2 (**B1**), and the corresponding images of total protein staining after membrane transfer used for normalization. The expression of enzymes involved in ceramide de novo synthesis pathway, i.e., SPTLC1 (**A2**) and SPTLC2 (**B2**) in primary rat hepatocytes. The cells were incubated with presence or absence of PA (0.5 mM) or/and COM (20 or 50 μM), as it was described in Materials and Methods section. The protein expression was evaluated by the Western blot technique. The data are shown as the mean ± SD and are based on six independent determinations (*n* = 6); * *p* < 0.05 significant difference vs. control group; # *p* < 0.05 significant difference vs. palmitate group; C, control; COM20 or COM50, coumestrol 20 or 50 μM; PA, palmitic acid; PA + COM20 or PA + COM50, palmitic acid + coumestrol 20 or 50 μM.

**Figure 4 biomolecules-11-00268-f004:**
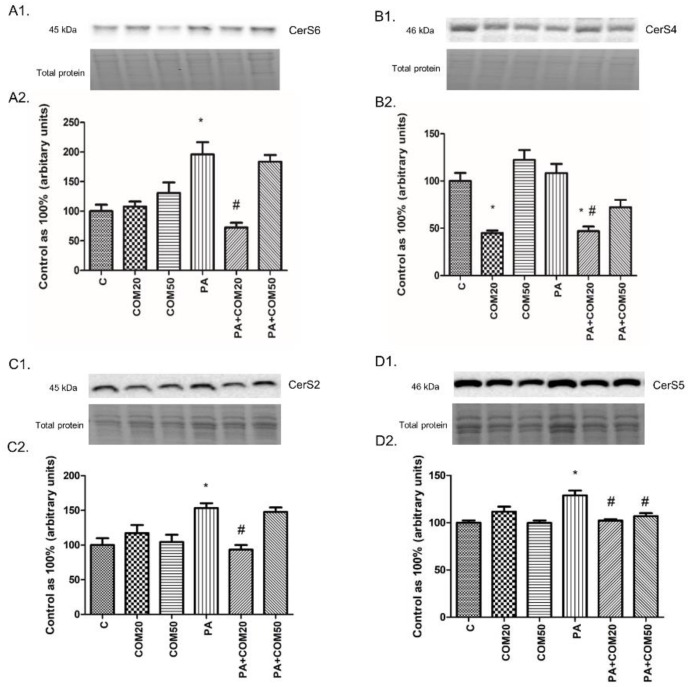
Representative Western Blot membranes of evaluated proteins, i.e., CerS6—ceramide synthase 6 (**A1**), CerS4—ceramide synthase 4 (**B1**), CerS2– ceramide synthase 2 (**C1**), CerS5– ceramide synthase 5 (**D1**), and the corresponding images of total protein staining after membrane transfer used for normalization. The expression of enzymes involved in both ceramide de novo synthesis and salvage pathways, i.e., CerS6 (**A2**), CerS4 (**B2**) CerS2 (**C2**), and CerS5 (**D2**) in primary rat hepatocytes. The cells were incubated with presence or absence of PA (0.5 mM) or/and COM (20 or 50 μM), as it was described in Materials and Methods section. The protein expression was evaluated by the Western blot technique. The data are shown as the mean ± SD and are based on six independent determinations (*n* = 6); * *p* < 0.05 significant difference vs. control group; # *p* < 0.05 significant difference vs. palmitate group; C, control; COM20 or COM50, coumestrol 20 or 50 μM; PA, palmitic acid; PA + COM20 or PA + COM50, palmitic acid + coumestrol 20 or 50 μM.

**Figure 5 biomolecules-11-00268-f005:**
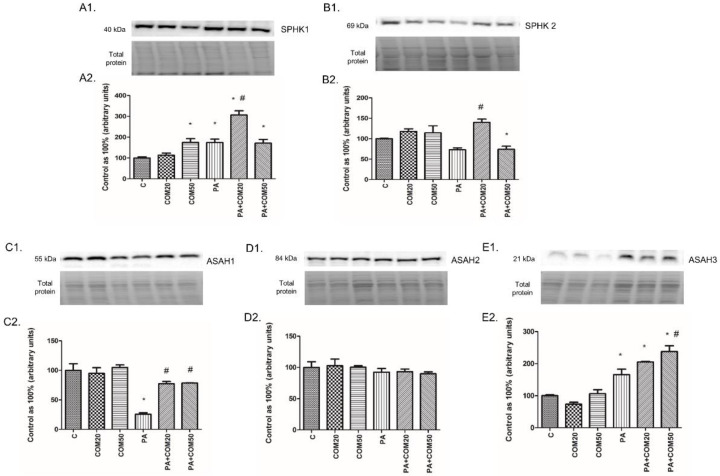
Representative Western blot membranes of evaluated proteins, i.e., SPHK1—sphingosine kinase 1 (**A1**), SPHK2 – sphingosine kinase 2 (**B1**), ASAH1—acid ceramidase (**C1**), ASAH2—neutral ceramidase (**D1**), ASAH3—alkaline ceramidase (**E1**), and the corresponding images of total protein staining after membrane transfer used for normalization. The expression of enzymes involved in sphingolipid salvage pathway, i.e., SPHK1 (**A2**), SPHK2 (**B2**), ASAH1 (**C2**), ASAH2 (**D2**), and ASAH3 (**E2**) in primary rat hepatocytes. The cells were incubated with presence or absence of PA (0.5 mM) or/and COM (20 or 50 μM), as it was described in Materials and Methods section. The protein expression was evaluated by the Western blot technique. The data are shown as the mean ± SD and are based on six independent determinations (*n* = 6); * *p* < 0.05 significant difference vs. control group; # *p* < 0.05 significant difference vs. palmitate group; C, control; COM20 or COM50, coumestrol 20 or 50 μM; PA, palmitic acid; PA + COM20 or PA + COM50, palmitic acid + coumestrol 20 or 50 μM.

**Figure 6 biomolecules-11-00268-f006:**
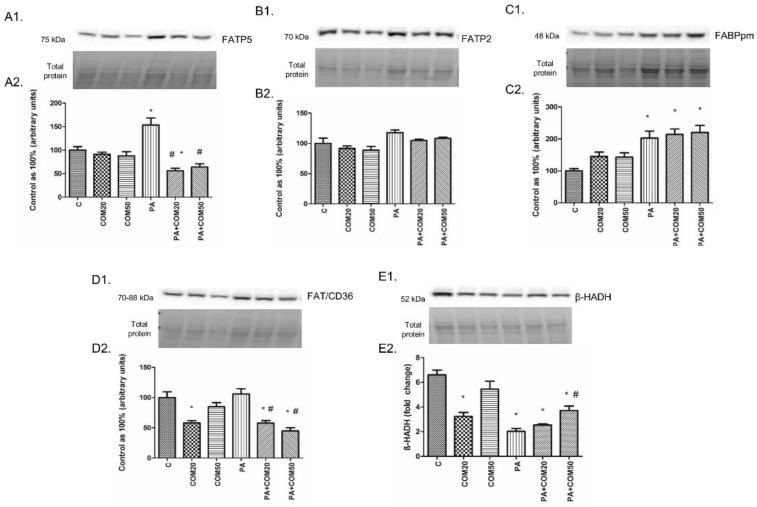
Representative Western blot membranes of evaluated proteins, i.e., FATP5 (**A1**), FATP2 (**B1**), FABPpm (**C1**), FAT/CD36 (**D1**), β-HADH (**E1**), and the corresponding images of total protein staining after membrane transfer used for normalization. The expression of proteins involved in fatty acid uptake, i.e., FATP5 (**A2**), FATP2 (**B2**), FABPpm (**C2**), FAT/CD36 (**D2**), and β-oxidation, namely, β-HADH (**E2**) in primary rat hepatocytes. The cells were incubated with presence or absence of PA (0.5 mM) or/and COM (20 or 50 μM), as it was described in Materials and Methods section. The protein expression was evaluated by the Western blot technique. The data are shown as the mean ± SD and are based on six independent determinations (*n* = 6); * *p* < 0.05 significant difference vs. control group; # *p* < 0.05 significant difference vs. palmitate group; C, control; COM20 or COM50, coumestrol 20 or 50 μM; PA, palmitic acid; PA + COM20 or PA + COM50, palmitic acid + coumestrol 20 or 50 μM.

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
