# Peer review of "The Influence of Coumestrol on Sphingolipid Signaling Pathway and Insulin Resistance Development in Primary Rat Hepatocytes"

_biomolecules, 2021, doi:10.3390/biom11020268_

Round 1

Reviewer 1 Report

The paper by Zywno et al. reports a study on the effects of the phytoestrogen coumestrol on sphingolipid biosynthesis/accumulation and signaling pathway as well as on insulin resistance parameters in a cell-based model: primary rat hepatocytes.

The authors report that coumestrol treatment resulted in a significant reduction of sphingolipid accumulation, with specific inhibition of ceramide de novo synthesis, increased the expression of FATP5 and FAT/CD36 and alterations of the sphingolipid salvage pathway.

The paper is clearly written and all sections are complete and understandable. Only minor revision/rephrasing of some lines are required.

Specific comments

Introduction

Please add a comment on the affinity of coumestrol to ERalpha and ERbeta and its potency compared to estradiol (to justify the concentrations used).

M&M

In section 2.2,

Line 96: is this a consolidated protocol? Can you add a reference for it?

Lines 100-102: please describe better when the insulin challenge was applied. Was it at the beginning or at the end of the 18h incubation with coumestrol?

Results

Fig. 1: COM20 treatment alone increased CER by over 51%: this is not commented in the Discussion. In any case, is this similar to what report by Charytoniuk (lines 318-319)?

Figures 2 to 5: representative WB plus a bar graph showing the densitometric analysis of 6 independent determinations are shown. I suggest to rename, for each parameter, as A1 the WB and A2 the related densitometry (or A and B and then continuing with the following letters) and consequently modify the legends.

Author Response

Bialystok, 24.01.2021

Dear Madam, Dear Sir,

We appreciate the time and effort that you dedicated to providing your valuable feedback on our manuscript entitled “The influence of coumestrol on the sphingolipid signaling pathway in primary rat hepatocytes”. We are grateful for your insightful comments on our paper. We have been able to incorporate changes to reflect most of the suggestions and concerns provided by the Reviewer. Furthermore, the whole manuscript was corrected and improved grammatically. The changes within the manuscript have been highlighted and all the references were adjusted for the new version of the manuscript.

Here are our point-by-point responses to the Reviewer’s comments and concerns:

Introduction

Please add a comment on the affinity of coumestrol to ERalpha and ERbeta and its potency compared to estradiol (to justify the concentrations used).

Authors: We added a brief comment describing the binding affinity and potency of COM in comparison with the other dietary phytoestrogens as well as endogenous 17β-estradiol (lines 53-55) supported by appropriate reference. (Hopert et al.).

M&M

In section 2.2,

Line 96: is this a consolidated protocol? Can you add a reference for it?

Authors: Yes, it is a consolidated protocol, as Reviewer suggested, we added an appropriate reference (Konstantynowicz-Nowicka et al.) (line 98).

Lines 100-102: please describe better when the insulin challenge was applied. Was it at the beginning or at the end of the 18h incubation with coumestrol?

Authors: We agree with the Reviewer that the moment of insulin application should be clarified. The cells were incubated with insulin for 15 min at the end of 18h incubation (lines 103-105).

Results

Fig. 1: COM20 treatment alone increased CER by over 51%: this is not commented in the Discussion. In any case, is this similar to what report by Charytoniuk (lines 318-319)?

Authors: We added a comment about the increased concentration of CER in the group incubated with COM20 alone as was suggested by the Reviewer. Furthermore, we linked this observation with the results after incubation with enterolactone alone what was described by Charytoniuk et al. and showed the discrepancy between our studies in the results after simultaneous incubation. In our study, we observed attenuated concentration of CER and SFA after incubation with both COM and PA (in two different concentrations) whereas, Charytoniuk et al. reported increased content of CER after incubation with both enterolactone and PA. Now, we hope that the issue is more cohesive (lines 347 and 349-354).

Figures 2 to 5: representative WB plus a bar graph showing the densitometric analysis of 6 independent determinations are shown. I suggest to rename, for each parameter, as A1 the WB and A2 the related densitometry (or A and B and then continuing with the following letters) and consequently modify the legends.

Authors: As it was suggested by the Reviewer, we renamed Figures 2 to 5. In each Figure appropriate letter and number 1 represents representative blot and number 2 represents a bar graph showing the densitometric analysis. Moreover, suitable information appeared in each Figure legend.

Taking into account all the Reviewer’s suggestions and concerns provided, we believe that the manuscript was further improved and is now more suitable for publication in the Biomolecules.

Yours faithfully,

Karolina Konstantynowicz-Nowicka

Reviewer 2 Report

Authors demonstrated that coumestrol: 1) affects palmitic acid-induced changes in sphingolipid amounts and the expression of the metabolic enzymes, β-HADH and fatty acid transport proteins, 2) ameliorates insulin resistance assessed by Akt and GSK phosphorylation in primary rat hepatocytes. Though the study is descriptive and mechanistic works are defect, the findings are important for understanding molecular mechanisms of coumestrol chemical biology and are believed to contribute the development of coumestrol-based chemoprevention and adjuvant therapy for metabolic disorders.

Major concerns are significant changes in COM20-treated samples shown in Figs 1A, 3B, 3D, 5D, and 5E. The data seem to spontaneous. Authors need to double-check the original data and confirm the reproductivity. If they are reproducible, authors should test the dose-dependency.

Other major points:

1) Sphingolipid metabolic pathway is complex and ceramide de novo pathway and salvage pathway share ceramide synthase (CerS). Authors need to carefully reorganize the definition of de novo and salvage pathway. Please carefully read the review article (PMID: 18191382).

2, Figs 3 and 4 need to be merged. In addition, the summary of changes can be shown as a new Table.

3, Significant changes shown in Figs 3 and 4 look very complicated. Authors need to improve the documentation and provide the discussion, very carefully.

4, Authors need to discuss the possible mechanisms by which ceramide decreases phosphorylation of AKT.

5, Authors need to interpret why just CerS4/6 among six CerS isoforms were assessed. Otherwise, CerS1/2/3/5 expression should be shown.

6, Figs 1A/C, in combination of PA and COM, COM dose-dependent downregulation of ceramide or SFA should be shown. Those results would strength the evidence.

Minor points  

1, LASS should be converted to CerS.

2, line 43-44, add the reference.

Author Response

Bialystok, 24.01.2021

Dear Madam, Dear Sir,

We appreciate the time and effort that you dedicated to providing your valuable feedback on our manuscript entitled “The influence of coumestrol on the sphingolipid signaling pathway in primary rat hepatocytes”. We are grateful for your insightful comments on our paper. We have been able to incorporate changes to reflect most of the suggestions and concerns provided by the Reviewer. Furthermore, the whole manuscript was corrected and improved grammatically. The changes within the manuscript have been highlighted and all the references were adjusted for the new version of the manuscript. 

Here are our point-by-point responses to the Reviewer’s comments and concerns:

Major concerns are significant changes in COM20-treated samples shown in Figs 1A, 3B, 3D, 5D, and 5E. The data seem to spontaneous. Authors need to double-check the original data and confirm the reproductivity. If they are reproducible, authors should test the dose-dependency.

Authors: As was suggested by the Reviewer we double-checked our data together with statistical analysis and we confirm that the results are reproducible. We selected these two coumestrol concentrations based on literature (Coumestrol modulates Akt and Wnt/β-catenin signaling during the attenuation of adipogenesis. Young Jin Jang, Hyo Jeong Son, Jiyun Ahn, Chang Hwa Junga and Taeyoul Ha; Food Funct. 2016 Dec 7;7(12):4984-4991) where scientists used concentrations 20, 40 and 60μM. They observed that coumestrol decreased lipid accumulation. Thus in our preliminary studies, we tested three coumestrol concentrations: 10, 20, 50μM, and 60μM. However, we did not observe any significant changes in tested parameters (sphingolipids as well as proteins) in the groups incubated with coumestrol 10μM alone and simultaneously with palmitate. Moreover, changes observed in groups incubated with 60μM of coumestrol showed the same changes in sphingolipid concentration as those incubated with 50μM. Thus we decided not to show these two concentrations because we did not want to distract readers’ attention. We added the information about the preliminary studies to the Materials and Methods section (line 100).

Other major points:

1) Sphingolipid metabolic pathway is complex and ceramide de novo pathway and salvage pathway share ceramide synthase (CerS). Authors need to carefully reorganize the definition of de novo and salvage pathway. Please carefully read the review article (PMID: 18191382).

Authors: Thank you for pointing this inconsistency. We are aware that ceramide synthases are present in both, de novo and salvage pathways as was pointed by the Reviewer. We also know that cellular homeostasis and availability of metabolites is a crucial factor determining which pathway is mainly activated. We incubated hepatocytes with palmitate that is a substrate for de novo ceramide synthesis and we assumed that this pathway must be activated the most. However, we completely agree with the Reviewer that the information in our manuscript seems to be confusing. Thus considering the article suggested by the Reviewer (Kitatani et al.), we:

(1) highlighted the importance of the sphingolipid salvage pathway in the overall ceramide formation (lines 370-372),

(2) added important possible correlation between de novo pathway and the salvage pathway. As the Reviewer pointed out, ceramide synthases share two pathways and the altered expression of these enzymes may affect both of them. Therefore, the strict separation of these different routes even in the figures should be avoided to ward off any ambiguities. We connected the decreased CerS expression with either de novo pathway and salvage pathway in contribution to the anti-diabetic effects of coumestrol (lines 393-403).

2, Figs 3 and 4 need to be merged. In addition, the summary of changes can be shown as a new Table.

Authors: We are afraid that connecting two figures may cause that especially blots will be illegible. Thus keeping in mind the previous Reviewer’s comment we decided to separate graphs depicting the expression of ceramide synthases. Thank you for your suggestion relating to the new table. However, we think that providing an extra table would be a simple repetition of the already existing graphs and would make our paper more challenging for reader.

3, Significant changes shown in Figs 3 and 4 look very complicated. Authors need to improve the documentation and provide the discussion, very carefully.

Authors: In the figures as well as in the description of the results we wanted to show all changes that were exerted by coumestrol, palmitate, and incubation with both substances. However, we agree with the Reviewer that some parts of the Results section may be improved. We removed the description of changes exerted by the PA group alone as it is not the main point of our manuscript (lines 214, 216, 218, 220, 222, 238, 240, 258-260, 262-267).

4, Authors need to discuss the possible mechanisms by which ceramide decreases phosphorylation of AKT.

Authors: We completely agree. As the Reviewer pointed out, we briefly described the possible mechanisms underlying the diminished Akt phosphorylation induced by ceramide which is crucial to the development of insulin resistance. We supported our statements with appropriate references (Sokolowska et al. and Bourbon et al.) (lines 327-331).

5, Authors need to interpret why just CerS4/6 among six CerS isoforms were assessed. Otherwise, CerS1/2/3/5 expression should be shown.

Authors: Due to the fact that ceramide synthase isoforms exert substance specificity and availability of different chain length acyl-CoA for incorporation and generation of ceramides with distinct acyl-chain lengths, in our experiment we incubated cells with palmitate (C16) thus we chose CerS4 (generates C18-ceramide  to C20-ceramide)[1] and CerS6 (generates mainly C16-ceramide)[2]. We added this information to the Discussion (line 401-402).

[1]Gault, C. R., Obeid, L. M., & Hannun, Y. A. (2010). An overview of sphingolipid metabolism: From synthesis to breakdown. Advances in Experimental Medicine and Biology, 688,1–23.

[2]Pewzner-Jung, Y., Ben-Dor, S., & Futerman, A. H. (2006). When do Lasses (longevity assurance genes) become CerS (ceramide synthases)?: Insights into the regulation of ceramide synthesis. The Journal of Biological Chemistry, 281(35), 25001–25005.

Moreover, being aware of already existing evidence, we decided to show the expression of only these two because they are considered as the most important in the obesity and insulin resistance development both in humans and animal models [3-5].

[3]. Sokolowska, E.; Blachnio-Zabielska, A. The Role of Ceramides in Insulin Resistance. Front. Endocrinol. (Lausanne). 2019, 10, 1–13.

[4].Timothy Hla, Richard Kolesnick, 16:0-Ceramide Signals Insulin Resistance, Cell Metabolism, Volume 20, Issue 5, 2014, Pages 703-705, ISSN 1550-4131.

[5].Suryaprakash Raichur, Bodo Brunner, Maximilian Bielohuby et al., The role of C16:0 ceramide in the development of obesity and type 2 diabetes: CerS6 inhibition as a novel therapeutic approach, Molecular Metabolism, Volume 21,2019, Pages 36-50.

6, Figs 1A/C, in combination of PA and COM, COM dose-dependent downregulation of ceramide or SFA should be shown. Those results would strength the evidence.

Authors: We agree with the Reviewer that such dose-dependent results would be attractive. However, as we explained at the beginning of the Reviewer’s comments we tested four coumestrol concentrations: 10, 20, 50μM, and 60μM. However, we did not observe any significant changes in tested parameters in the groups incubated with coumestrol 10μM alone and simultaneously with palmitate. The changes in sphingolipid concentration observed in groups incubated with 60μM of coumestrol were the same as these incubated with 50μM. Thus changes in our experiment were not dose-dependent in primary hepatocytes what is contradictory to the effects exerted in adipocytes (Coumestrol modulates Akt and Wnt/β-catenin signaling during the attenuation of adipogenesis. Young Jin Jang, Hyo Jeong Son, Jiyun Ahn, Chang Hwa Junga and Taeyoul Ha; Food Funct. 2016 Dec 7;7(12):4984-4991).

Minor points  

1, LASS should be converted to CerS.

Authors: We have converted the abbreviation of ceramide synthase from LASS to CerS, consequently within the whole manuscript.

2, line 43-44, add the reference.

Authors: We added appropriate reference suggested by the Reviewer (Kitatani et al.) (line 44).

Taking into account all the Reviewer’s suggestions and concerns provided, we believe that the manuscript was further improved and is now more suitable for publication in the Biomolecules.

Yours faithfully,

Karolina Konstantynowicz-Nowicka

Round 2

Reviewer 2 Report

Still some major concerns remain.

1) Authors should show the data as to the COM-dose responses (10, 20, 50μM, and 60μM COM in the presence of palmitates).

2) Authors should give comments about the significant changes : COM20 alone significantly increased CER (Fig 1A), COM20/50 without palmitatessignificantly decreased S1P (Fig 1D).

3) Palmitates are incorporated into cells and converted to palmitoyl-CoA that can be a substrate for SPT as well as predominantly CerS5/6. Substantially, palmitate treatment was shown to increase multiple ceramide species (PMID: 19369694 ). So, authors need to adequately interpret why just CerS4/6 among six CerS isoforms were assessed. Otherwise, CerS1/2/3/5 expression should be shown.

Author Response

Bialystok, 09.02.2021

Dear Madam, Dear Sir,

Thank You for Your further revision of our manuscript. We are grateful for your in-depth comments on our paper. We have been able to incorporate changes to reflect most of the concerns provided by the Reviewer. The changes within the manuscript have been highlighted. 

Here are our point-by-point responses to the Reviewer’s comments and concerns:

1) Authors should show the data as to the COM-dose responses (10, 20, 50μM, and 60μM COM in the presence of palmitates).

Authors: We agree that such results may be important. As it was suggested by the Reviewer, dose-dependent changes in sphingolipids concentrations were added as supplementary material in order not to overload the readers with too much data (line 103).

2) Authors should give comments about the significant changes : COM20 alone significantly increased CER (Fig 1A), COM20/50 without palmitates significantly decreased S1P (Fig 1D).

Authors: Thank You for Your elucidation.  As the Reviewer pointed out, we implemented comments describing significant changes in COM20 treated groups both for CER and S1P (lines 163-164, 173-176 and 352).

3) Palmitates are incorporated into cells and converted to palmitoyl-CoA that can be a substrate for SPT as well as predominantly CerS5/6. Substantially, palmitate treatment was shown to increase multiple ceramide species (PMID: 19369694 ). So, authors need to adequately interpret why just CerS4/6 among six CerS isoforms were assessed. Otherwise, CerS1/2/3/5 expression should be shown.

Authors: As it was suggested by the Reviewer, we examined the expression of additional isoforms of the ceramide synthase namely CerS2 and CerS5. The significant changes which we observed remain in agreement with the changes in the expression of CerS4 and CerS6 described in our study, suggesting the possible involvement of different CerS isoforms in the altered ceramide synthesis induced by coumestrol. Moreover, we tested also the expression of CerS1 and CerS3. However, we did not receive any results (the bands on the blot were not observed at all) probably because these isoforms are expressed in different tissues namely the brain and skeletal muscles (CerS1) as well as skin and testis (CerS3) not in the liver/hepatocytes [1,2]. Furthermore, as we mentioned before, alterations in CerS2/4/5/6 expression are positively correlated with the development of the insulin resistance and obesity [3]. Taking into account the Reviewer suggestions we believe that providing more CerS isoforms will now comprehensively show the link between attenuated induced by coumestrol expression of these enzymes and amelioration of the insulin sensitivity further described in our manuscript. We added appropriate comments in the M&M, results, and discussion, as well as an updated figure 4 (lines 117-118, 238-249, 368, 399, 407, Fig 4.)  

[1]Takayuki Sassa, Akio Kihara. Metabolism of very long-chain Fatty acids: genes and pathophysiology. Biomol Ther (Seoul). 2014 Feb;22(2):83-92. doi: 10.4062/biomolther.2014.017.

[2]. Grösch S, Schiffmann S, Geisslinger G. Chain length-specific properties of ceramides. Prog Lipid Res. 2012 Jan;51(1):50-62. doi: 10.1016/j.plipres.2011.11.001. Epub 2011 Nov 25. PMID: 22133871.

[3].Hla T, Kolesnick R. C16:0-ceramide signals insulin resistance. Cell Metab. 2014;20(5):703-705. doi:10.1016/j.cmet.2014.10.017

Taking into account all the Reviewer’s suggestions and concerns provided, we believe that the manuscript was further improved and is now more suitable for publication in the Biomolecules.

Yours faithfully,

Karolina Konstantynowicz-Nowicka
